# Troponin in COVID-19: To Measure or Not to Measure? Insights from a Prospective Cohort Study

**DOI:** 10.3390/jcm11195951

**Published:** 2022-10-09

**Authors:** Renata Moll-Bernardes, João D. Mattos, Eduardo B. Schaustz, Andréa S. Sousa, Juliana R. Ferreira, Mariana B. Tortelly, Adriana M. L. Pimentel, Ana Cristina B. S. Figueiredo, Marcia M. Noya-Rabelo, Allan R. K. Sales, Denilson C. Albuquerque, Paulo H. Rosado-de-Castro, Gabriel C. Camargo, Olga F. Souza, Fernando A. Bozza, Emiliano Medei, Ronir R. Luiz

**Affiliations:** 1D’Or Institute for Research and Education, Rio de Janeiro 22281-100, Brazil; 2Evandro Chagas National Institute of Infectious Disease, Oswaldo Cruz Foundation, Rio de Janeiro 21040-360, Brazil; 3Cardiology Unit, Copa D’Or Hospital, Rio de Janeiro 22031-011, Brazil; 4Cardiology Unit, Niterói D’Or Hospital, Rio de Janeiro 24230-251, Brazil; 5Cardiology Unit, Glória D’Or Hospital, Rio de Janeiro 22211-230, Brazil; 6Cardiology Department, Bahia School of Medicine and Public Health, Salvador 40290-000, Brazil; 7Cardiology Unit, Aliança Hospital, Salvador 41920-180, Brazil; 8Cardiology Department, Rio de Janeiro State University, Rio de Janeiro 20551-030, Brazil; 9National Center for Structural Biology and Bioimaging, Federal University of Rio de Janeiro, Rio de Janeiro 21941-902, Brazil; 10Institute for Studies in Public Health—IESC, Federal University of Rio de Janeiro, Rio de Janeiro 21941-592, Brazil

**Keywords:** troponin, myocardial injury, biomarker, COVID-19, prognosis

## Abstract

Myocardial injury (MI), defined by troponin elevation, has been associated with increased mortality and adverse outcomes in patients with coronavirus disease 2019 (COVID-19), but the role of this biomarker as a risk predictor remains unclear. Data from adult patients hospitalized with COVID-19 were recorded prospectively. A multiple logistic regression model was used to quantify associations of all variables with in-hospital mortality, including the calculation of odds ratios (ORs) and confidence intervals (CI). Troponin measurement was performed in 1476 of 4628 included patients, and MI was detected in 353 patients, with a prevalence of 23.9%; [95% CI, 21.8–26.1%]. The total in-hospital mortality rate was 10.9% [95% CI, 9.8–12.0%]. The mortality was much higher among patients with MI than among those without MI, with a prevalence of 22.7% [95% CI, 18.5–27.3%] vs. 5.5% [95% CI, 4.3–7.0%] and increased with each troponin level. After adjustment for age and comorbidities, the model revealed that the mortality risk was greater for patients with MI [OR = 2.99; 95% CI, 2.06–4.36%], and for those who did not undergo troponin measurement [OR = 2.2; 95% CI, 1.62–2.97%], compared to those without MI. Our data support the role of troponin as an important risk predictor for these patients, capable of discriminating between those with a low or increased mortality rate. In addition, our findings suggest that this biomarker has a remarkable negative predictive value in COVID-19.

## 1. Introduction

Myocardial injury, defined by cardiac troponin elevation, is a common finding in patients with coronavirus disease 2019 (COVID-19), with a prevalence among hospitalized patients ranging from 7% to 44% [1,2,3,4,5]. It has been associated with adverse outcomes and mortality, but gaps remain in our understanding of its epidemiology and clinical implications [6].

The pathogenesis of myocardial injury in COVID-19 is not completely understood but is likely multifactorial. According to the Fourth Universal Definition of Myocardial Infarction [7], myocardial injury may be present in cases of acute myocardial ischemia due to plaque rupture associated with atherosclerosis and thrombosis (type 1 infarction), or due to an imbalance between oxygen supply and demand, which can occur in the presence of tachyarrhythmias or in critically ill patients (type 2 infarction). Myocardial injury can also be associated with chronic non-ischemic injuries, such as chronic kidney disease and heart failure, as well as with acute non-ischemic injuries, such as myocarditis, direct virus-mediated injury, inflammation caused by immune response exacerbation, and endothelial dysfunction [7].

In the context of COVID-19, myocardial injury is more likely to occur in older patients, those with chronic cardiovascular conditions, and those with severe presentations [2,3]. The increased risk of myocardial injury in patients with cardiovascular comorbidities is probably related to a dysregulated immune response and a greater angiotensin-converting enzyme 2 expression [8,9,10,11,12]. This enzyme, which is expressed in many organs, such as the heart, lungs, and kidneys, is a key regulator of the renin-angiotensin-aldosterone system, and a cell-surface functional receptor for severe acute respiratory syndrome coronavirus 2 (SARS-CoV-2) [13,14].

Despite the results presented in previous studies, including a meta-analysis that revealed clinically meaningful relationships between troponin levels and COVID-19 severity and mortality [1], controversy remains; a few authors have argued that troponin measurement provides limited prognostic information in this context, as levels reflect baseline comorbidities, advanced age, respiratory function, and multisystem organ dysfunction [6,15,16]. Due to the lack of definitive evidence regarding the role of troponin measurement, American and European cardiology societies do not recommend its routine performance to increase risk stratification for patients hospitalized with COVID-19 [17,18,19]. In this study, we further explored the value of routinely measuring troponin to predict the risk of death and adverse outcomes in a large multicentric population of patients hospitalized with COVID-19.

## 2. Materials and Methods

### 2.1. Population and Design

Consecutive adult patients hospitalized with suspected or confirmed COVID-19 diagnoses in 35 tertiary hospitals in Brazil were included prospectively in a multicenter registry. The SARS-CoV-2 infection was confirmed by real-time reverse-transcription polymerase chain reaction of nasopharyngeal and/or oropharyngeal swab samples. The study was approved by the Brazilian Ministry of Health’s National Commission for Research Ethics and the Institutional Review Boards or Ethics Committees of the participating sites (CAAE#29496920.8.0000.5262) and, due to the minimal risk of this observational study and limited access to patients during the pandemic, informed consent was waived.

Trained investigators collected demographic, clinical, and laboratory data using the standardized form from the International Severe Acute Respiratory and Emerging Infection Consortium/World Health Organization Clinical Characterization Protocol (ISARIC/WHO) [20]. Data were collected from electronic medical records and entered into electronic case-report forms using Research Electronic Data Capture (RedCap) platform (Vanderbilt University, Nashville, TN, USA). Clinical data included: comorbidities, symptoms, and vital signs at the time of hospital admission; complications; and treatment. Laboratory tests were performed throughout hospitalization according to local clinical practice. The patients were followed prospectively until hospital discharge or in-hospital death. They were categorized according to age (<40, 40–59, 60–69, and ≥70 years).

To enable the comparison of data obtained with different troponin assays, we normalized values to the 99th percentile upper reference limit (URL) of each assay (Appendix A) and recorded the results as ratios. We then categorized troponin ratios as normal (≤1× URL) and mildly (>1 to ≤3× URL), moderately (>3 to ≤10× URL), and severely (>10× URL) elevated, similar to the categorization used by Majure et al. [1]. Myocardial injury was defined as troponin value >1× URL, measured in the first week of hospitalization to avoid the influence of confounders such as secondary bacterial infection and sepsis. For patients who had more than one troponin measurement in the first week we considered the peak value. Patients with no troponin measurement were included in our multiple logistic regression analysis as an additional troponin category.

### 2.2. Outcomes

The primary outcome was in-hospital mortality in patients hospitalized with COVID-19. The secondary outcomes included complications such as thromboembolic phenomena (including deep venous thrombosis and pulmonary embolism), stroke, myopericarditis, heart failure, acute renal failure, and sepsis. The outcomes were reported according to troponin measurement and troponin levels in the first week of hospitalization.

### 2.3. Statistical Analysis

Continuous variables were categorized, and categorical variables were characterized by proportions. The chi-squared test was used to detect associations between degrees of myocardial injury and clinical variables; significance was defined as *p* ≤ 0.05. A myocardial injury prevalence ratio was calculated for each categorical variable. The chi-squared test was also used to compare the incidence of complications among troponin categories. For the primary outcome, a confidence interval (CI) and relative risk (RR) value were calculated for each clinical variable. A multiple logistic regression model was used to quantify associations of all variables with the primary outcome, with the calculation of unadjusted and adjusted odds ratios (ORs) and CIs. The goodness of fit of the final model was evaluated using the Hosmer–Lemeshow test. Predicted probabilities for the primary outcome were estimated using variables showing significant associations in the final model. All analyses were performed using SPSS software (version 24.0; IBM Corporation, Armonk, NY, USA).

## 3. Results

During the recruitment period (17 March 2020–5 July 2021), data from 4628 patients were included in the COVID-19 registry. Patients with incomplete clinical and laboratory data or no confirmation of SARS-CoV-2 infection (*n* = 1382) were excluded from the analysis (Appendix A). From 3246 patients included in this cohort study, 18.6% were aged <40 years, 41.8% were aged 40–59 years, 16.7% were aged 60–69 years, and 23% were aged ≥70 years (23%); 1311 (40.4%) patients were female, 1843 (56.8%) were hypertensive, and 812 (25%) had diabetes. Heart failure was reported in 101 (3.1%) patients, and 217 (6.7%) patients had coronary artery disease. Chronic cardiac and kidney disease were reported in 167 (5.1%) and 152 (4.7%) cases, respectively (Table 1). Troponin measurement was performed in the first week of hospitalization in 1476 (45.5%) patients, and myocardial injury was detected in 353 patients with a prevalence of 23.9% [95% CI, 21.8–26.1%]. Troponin levels were mildly elevated in 127 (8.6%) patients, moderately elevated in 87 (5.9%), and severely elevated in 139 (9.4%) patients (Figure 1).

Myocardial injury was more prevalent in older patients with prevalence of (46.2% (≥70 years), 33.1% (60–69 years), 14.0% (40–59 years), and 7.5% (<40 years)) and in those with lower oxygen saturation (30.9% vs. 21.1%), diabetes (31.9% vs. 21.2%), hypertension (30.7% vs. 14.3%), heart failure (58.3% vs. 22.8%), coronary artery disease (44.2% vs. 22.2%), chronic cardiac disease (47.8% vs. 22.3%), and chronic kidney disease (50.0% vs. 22.8%; all *p* < 0.001; Table 2). The prevalence of myocardial injury did not differ according to sex. The proportions of patients with comorbidities are provided by troponin category in Appendix A.

The groups that did and did not undergo troponin measurement were similar with regard to the sex distribution (*p* = 0.28), age (*p* = 0.14), and the frequencies of comorbidities such as reduced oxygen saturation (*p* = 0.81), diabetes (*p* = 0.76), hypertension (*p* = 0.09), heart failure (*p* = 0.67), chronic renal disease (*p* = 0.06), stroke or transient ischemic attack (*p* = 0.50), and acute renal failure (*p* = 0.16; Table 1).

### 3.1. Primary Outcome—In-Hospital Mortality

The overall in-hospital mortality rate was 10.9% [95% CI, 9.8–12.0%]. In-hospital mortality was associated with older age (24.8% for ≥70 years vs. 2.5% for <40 years; RRs = 2.5 (40–59 years), 5.0 (60–69 years), and 9.9 (≥70 years)), oxygen saturation ≤93% (RR = 2.3), diabetes (RR = 1.9), hypertension (RR = 1.7), heart failure (RR = 2.4), coronary artery disease (RR = 1.7), chronic heart disease (RR = 2.0), and chronic kidney disease (RR = 2.5; all *p* < 0.001; Table 3). It was not associated with patient sex.

The mortality rate was slightly higher for patients who did not undergo troponin measurement than for those who did, 11.9% [95% CI, 10.5–13.5%] vs. 9.6% [95% CI, 8.2–11.2%; *p* = 0.04; Figure 2A]. Interestingly, in the groups of patients for whom troponin was measured, the mortality was markedly higher for patients with than for those without cardiac injury, 22.7% [95% CI, 18.5–27.3%]) vs. 5.5% [95% CI, 4.3–7.0%]. Additionally, it was increased by troponin level from 5.5% (≤1× URL—no cardiac injury) to 13.4% (>1 to ≤3× URL), 23.0% (>3 to ≤10× URL), and 30.9% (>10× URL; *p* < 0.001; Figure 2B; Appendix A).

### 3.2. Secondary Outcomes

Several complications were detected during hospitalization and were associated with in-hospital mortality as evidenced by an increased RR of death for each adverse outcome. Thromboembolic phenomena were detected in 120 (3.7%) patients (RR = 2.5), and stroke or transient ischemic attack (TIA) occurred in 92 (2.8%) patients (RR = 1.7). Myopericarditis was detected in 45 (1.4%) patients (RR = 2.3), heart failure in 82 (2.5%) patients, and myocardial ischemia in 72 (2.2%) patients (RR = 3.2). Sepsis or septic shock was evidenced in 339 (10.4%) cases (RR = 9.3), acute renal failure occurred in 380 (11.7%) cases (RR = 12.3), and in 621 (19.1%) cases the patients required invasive ventilatory support (RR = 36.4; Appendix A).

The incidences of thromboembolic phenomena, stroke or TIA, heart failure, and acute renal failure were similar in the patients who did and did not undergo troponin measurement in the first week of hospitalization. As expected, myopericarditis and myocardial ischemia were more frequent among patients who underwent troponin measurement (2.0% vs. 0.8%; *p* = 0.004 and 3.3% vs. 1.4%; *p* < 0.001, respectively). Sepsis and septic shock were also more common in patients who had undergone troponin measurement (11.8% vs. 9.3%; *p* = 0.02). Respiratory failure requiring invasive ventilatory support was more frequent among patients who did not undergo troponin measurement (21% vs. 16.9%; *p* = 0.004; Appendix A; Figure 3A).

As expected, myocarditis and myocardial ischemia were associated with myocardial injury (4.8% vs. 1.2% and 9.1% vs. 1.4%, respectively; both *p* < 0.001). Stroke or TIA were not associated with myocardial injury (4.0% and 2.8%; *p* = 0.39). Other complications were more frequent in patients with myocardial injury (thromboembolic phenomena; 4.0% vs. 2.6%, *p* = 0.047; heart failure; 6.5% vs. 1.3%, *p* < 0.001; sepsis or septic shock; 20.1% vs. 9.2%, *p* < 0.001; acute renal failure; 21.8% vs. 7.4%, *p* < 0.001; respiratory failure requiring mechanical ventilation; 36.5% vs. 10.8%, *p* < 0.001; Figure 3B. In addition, among the patients with myocardial injury, the frequency of invasive ventilation, acute renal failure, sepsis, and septic shock increased with each troponin level (Figure 4).

Troponin measurement was more frequent in the presence than in the absence of coronary artery disease (52.1% vs. 45%; p = 0.04), chronic cardiac disease (55.1% vs. 44.9%; *p* = 0.01), myopericarditis (66.7% vs. 45.2%; *p* = 0.004), myocardial ischemia (66.7% vs. 45.0%; *p* < 0.001), and sepsis or septic shock (51.3% vs. 44.8%; *p* = 0.02). Troponin measurement was less frequent in the presence than in the absence of thromboembolic phenomena (35.8% vs. 45.8%; *p* = 0.03), respiratory failure leading to mechanical ventilation (40.3% vs. 46.7%; *p* = 0.004) and death (40.2% vs. 46.1%; *p* = 0.04).

### 3.3. Multivariate Analysis Results

After adjustment for age, preexisting cardiovascular disease (hypertension, coronary heart disease, heart failure, and chronic heart disease), diabetes mellitus, chronic renal disease, and troponin level (normal, elevated, or not measured), the multivariable logistic regression model to predict death revealed a significantly greater risk in patients with cardiac injury [OR = 2.99; 95% CI, 2.06–4.36%]. This risk was also elevated in patients who did not undergo troponin measurement relative to those who had normal troponin levels [adjusted OR = 2.2; 95% CI, 1.62–2.97%], highlighting the high negative predictive value of troponin. Age was another independent risk factor for mortality with COVID-19 (ORs = 2.32 (40–59 years), 4.12 (60–69 years), and 9.07 (≥70 years), relative to <40 years). Other independent risk factors for mortality were oxygen saturation ≤93% [OR = 2.17; 95% CI, 1.71–2.75%] and diabetes [OR = 1.35; 95% CI, 1.05–1.73%]. The other comorbidities did not maintain associations with mortality when combined with troponin assessment (Table 4).

### 3.4. Predictive Model

The initial model for the prediction of mortality in patients hospitalized with COVID-19 included the troponin level, sex, age, oxygen saturation, diabetes, hypertension, heart failure, coronary artery disease, chronic heart disease, and chronic renal disease. After the exclusion of variables not associated with mortality in the multivariate analysis (*p* > 0.05), the final predictive model included myocardial injury, age, diabetes, and oxygen saturation. The risk of in-hospital mortality ranged from 1.2% (patients less than 40 years old, with troponin level ≤1× URL, no diabetes and oxygen saturation > 93%) to 48.2% (patients 70 years and older with diabetes, troponin level >1× URL and oxygen saturation <93%; Table 5).

## 4. Discussion

In the present study, troponin levels did allow for the discrimination of a group with no myocardial injury and a low mortality rate from a group with myocardial injury and increased mortality. In addition, the mortality rate increased with each troponin level, highlighting the role of troponin as an important risk predictor. Although the mortality rate differed slightly between groups that did and did not undergo troponin measurement in the first week of hospitalization, the patients who lacked troponin measurement had an intermediate risk relative to the patients with and without myocardial injury, suggesting that this biomarker has remarkable negative predictive value in the context of COVID-19.

In agreement with our findings, Chapman et al. [21] argued against the avoidance of troponin testing, suggesting that we must gain a better understanding of the usefulness of this biomarker, which could have important prognostic value, and that clinicians must be educated about the interpretation and implications of troponin elevation in patients with COVID-19.

Additionally, we analyzed the prevalence of myocardial injury and in-hospital mortality according to the baseline characteristics of patients hospitalized with COVID-19. Myocardial injury was detected via troponin elevation during the first week of hospitalization in 23.9% of patients who underwent such measurement. The in-hospital mortality rate was higher in patients with cardiac injury and increased with the troponin level. In-hospital mortality was also associated with age, oxygen saturation, and all cardiovascular comorbidities, but not sex. Similar to our findings, Shi et al. [2] reported that cardiac injury was a common complication (19.7%) associated with a greater risk of in-hospital mortality in 416 patients hospitalized with COVID-19 in Wuhan, China. Guo et al. [3] demonstrated that patients with COVID-19 and myocardial injury were older and had a greater prevalence of chronic cardiovascular conditions.

Myocardial injury in patients with COVID-19 may be related to various mechanisms, including primary ischemic disease, an oxygen supply/demand imbalance due to hypoxemia and reduced myocardial oxygen supply, and non-ischemic causes, such as myocarditis and stress cardiomyopathy [22]. Patients with COVID-19 are at increased risk of ischemia and myocardial injury due to increased sympathetic activation, hypercoagulability, a dysregulated immune response, vasculitis, and potential direct myocardial infection [2,23]. These mechanisms may explain our finding of associations of multiple complications assessed in this study with the presence of myocardial injury. The high sensitivity of troponin assays may permit early suspicion of these potential complications, enabling early treatment and improving outcomes.

Based on an analysis of data from 243 patients with COVID-19, Metkus et al. [6] reported that myocardial injury in the context of severe COVID-19 was a function of baseline comorbidities, advanced age, and multisystem organ dysfunction; they observed no association between myocardial injury and mortality in an analysis adjusted for age, sex, and multisystem organ dysfunction. In contrast, our multivariable logistic regression model showed that cardiac injury was an independent predictor of mortality in patients with COVID-19. In addition, patients who did not undergo troponin measurement were at increased risk of in-hospital mortality relative to patients with normal troponin levels. Age, oxygen saturation ≤93%, and diabetes were also independent risk factors for mortality in patients with COVID-19 in our model. The discrepancies in findings are likely related to differences in the study populations; Metkus et al. (2021) included only patients with severe and critical (requiring intubation) COVID-19 in their sample. In addition, they reported that troponin levels were assessed 24 h after intubation, reflecting a later stage of the disease relative to the conditions of the patients in our study.

Our final predictive model for in-hospital mortality included myocardial injury, age, diabetes, and oxygen saturation. We included patients with no troponin measurement in the analysis as a third category, so all 3246 patients were incorporated in the model. Although uncommonly used, this strategy is more comprehensive, permitting an adjustment for other covariates such as age, sex and comorbidities, and avoiding any selection bias that could arise if we had excluded these patients.

Troponin measurement during the first week of hospitalization may aid the identification of patients at greater risk of in-hospital death and cardiovascular complications, who must be closely surveilled, as they will probably need intensive care or other invasive support. A negative troponin result may indicate a better prognosis and a higher probability that a patient will not need intensive support and could have an earlier hospital discharge. Patients with no troponin measurement might have a better risk prediction if we measured this biomarker, as the lack of measurement was associated with an OR of 2.2, compared to patients with no myocardial injury, after adjusting for age, sex, oxygen saturation, and other comorbidities. Measuring troponin in the patients hospitalized with COVID-19 would provide an improved patient care and resource allocation. Nevertheless, clinicians need to better understand the role of this biomarker to avoid the performance of inappropriate diagnostic and therapeutic procedures.

To our knowledge, this was the first study to consider the lack of troponin measurement in the analysis to assess the prognostic value of this biomarker in COVID-19. Our findings show that troponin is an independent risk predictor and we suggest that troponin should be routinely measured in patients hospitalized with COVID-19.

This study has some limitations. Due to its observational nature, troponin measurement was not performed for all of the included patients, reflecting current clinical practice. We did not include obesity in our analysis due to lack of good quality information about this comorbidity. In addition, a substantial number of patients were excluded from the analysis because some centers did not have enough research personal to collect clinical data during some periods of the pandemic; however, the missing data pattern was completely at random, and unrelated to comorbidities, outcomes, or to the presence of myocardial injury. Overall, the large number of patients included and the multicentric nature of the study provided robust internal and external validity. Further studies are necessary to improve the understanding of the pathophysiological mechanisms of myocardial injury associated with SARS-CoV2 infection, and long-term evolution of survivors.

## 5. Conclusions

Myocardial injury was associated with age, low oxygen saturation, and cardiovascular comorbidities, with a prevalence of 23.9% in the present sample of patients hospitalized with COVID-19. In addition, higher levels of troponin were associated with increased in-hospital mortality and the most adverse cardiovascular outcomes. Patients who did not undergo troponin measurement had an intermediate risk of death relative to patients with and without myocardial injury, and the absence of myocardial injury was associated with a better prognosis. These findings suggest that troponin should be measured in patients hospitalized with COVID-19, as it is a useful biomarker that may improve risk prediction and resource utilization in this context. Educational measures to enhance clinicians’ understanding of the role of troponin as a biomarker in the context of COVID-19 are recommended.

## Figures and Tables

**Figure 1 jcm-11-05951-f001:**
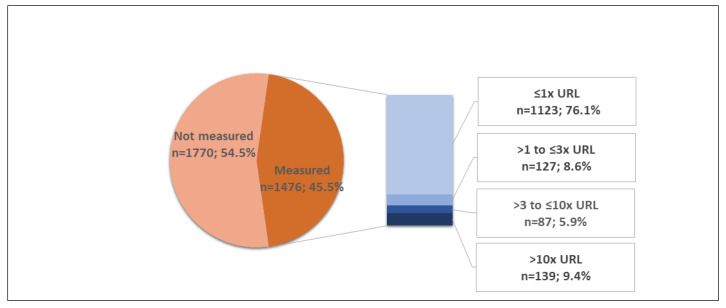
Prevalence of myocardial injury during the first week of hospitalization. Troponin ratios were categorized as normal (≤1× URL) and mildly (>1 to ≤3× URL), moderately (>3 to ≤10× URL), and severely (>10× URL) elevated. URL, upper reference limit.

**Figure 2 jcm-11-05951-f002:**
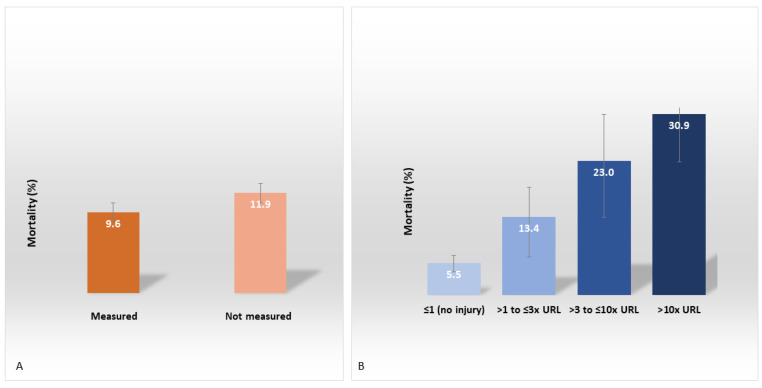
In-hospital mortality among patients with COVID-19 according to (**A**) troponin measurement (*n* = 3246) and (**B**) troponin levels (*n* = 1476) in the first week of hospitalization.

**Figure 3 jcm-11-05951-f003:**
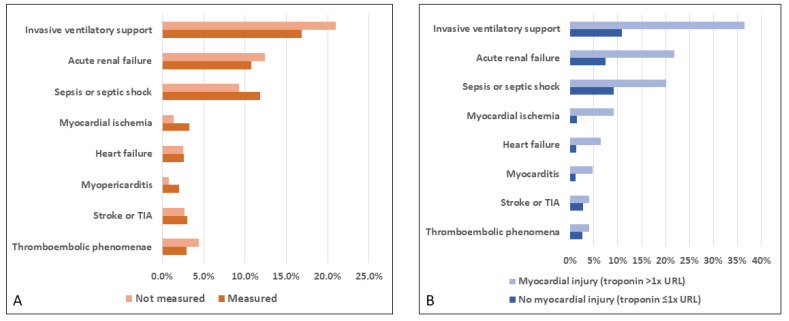
Complications according to (**A**) troponin measurement (*n* = 3246) and (**B**) myocardial injury (*n* = 1476) in the first week of hospitalization. TIA, transient ischemic attack; URL, upper reference limit.

**Figure 4 jcm-11-05951-f004:**
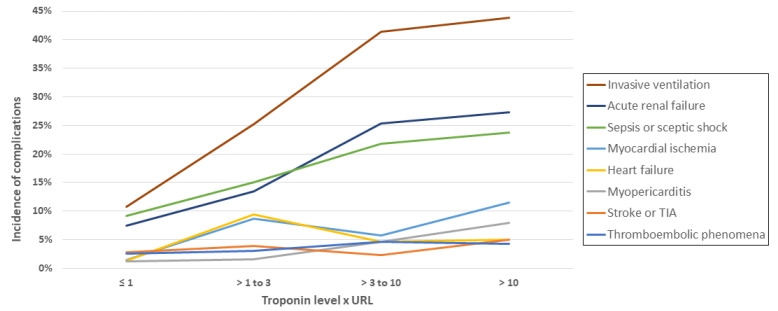
Incidence of in-hospital complications according to troponin level (*n* = 1476). TIA, transient ischemic attack; URL, upper reference limit.

**Table 1 jcm-11-05951-t001:** Baseline patient characteristics according to troponin measurement.

Characteristics	Total		Measured Troponin				χ^2^ Test *p*-Value
			No		Yes		
	*n*	(%)	*n*	(%)	*n*	(%)	
**All patients**	3246	(100.0)	1770	(54.5)	1476	(45.5)	
**Sex**							
Male	1935	(59.6)	1040	(53.7)	895	(46.3)	0.277
Female	1311	(40.4)	730	(55.7)	581	(44.3)	
**Age (years)**							
<40	603	(18.6)	349	(57.9)	254	(42.1)	0.140
40–59	1357	(41.8)	748	(55.1)	609	(44.9)	
60–69	541	(16.7)	281	(51.9)	260	(48.1)	
≥70	745	(23.0)	392	(52.6)	353	(47.4)	
**O_2_ saturation**							
>93%	2327	(71.7)	1272	(54.7)	1055	(45.3)	0.807
≤93%	919	(28.3)	498	(54.2)	421	(45.8)	
**Diabetes**							
No	2434	(75.0)	1331	(54.7)	1103	(45.3)	0.759
Yes	812	(25.0)	439	(54.1)	373	(45.9)	
**Hypertension**							
No	1403	(43.2)	789	(56.2)	614	(43.8)	0.088
Yes	1843	(56.8)	981	(53.2)	862	(46.8)	
**Heart failure**							
No	3145	(96.9)	1717	(54.6)	1428	(45.4)	0.674
Yes	101	(3.1)	53	(52.5)	48	(47.5)	
**Coronary disease**							
No	3029	(93.3)	1666	(55.0)	1363	(45.0)	0.043
Yes	217	(6.7)	104	(47.9)	113	(52.1)	
**Chronic heart disease**							
No	3079	(94.9)	1695	(55.1)	1384	(44.9)	0.010
Yes	167	(5.1)	75	(44.9)	92	(55.1)	
**Chronic Kidney disease**							
No	3094	(95.3)	1676	(54.2)	1418	(45.8)	0.064
Yes	152	(4.7)	94	(61.8)	58	(38.2)	
**Thromboembolic phenomena ***							
No	3126	(96.3)	1693	(54.2)	1433	(45.8)	0.031
Yes	120	(3.7)	77	(64.2)	43	(35.8)	
**Stroke or TIA**							
No	3154	(97.2)	1723	(54.6)	1431	(45.4)	0.501
Yes	92	(2.8)	47	(51.1)	45	(48.9)	
**Myopericarditis**							
No	3201	(98.6)	1755	(54.8)	1446	(45.2)	0.004
Yes	45	(1.4)	15	(33.3)	30	(66.7)	
**Myocardial ischemia ****							
No	3174	(97)	1746	(55.0)	1428	(45.0)	<0.001
Yes	72	(3.0)	24	(33.3)	48	(66.7)	
**Acute renal failure**							
No	2866	(88.3)	1550	(54.1)	1316	(45.9)	0.161
Yes	380	(11.7)	220	(57.9)	160	(42.1)	
**Sepsis or septic shock**							
No	2907	(89.6)	1605	(55.2)	1302	(44.8)	0.022
Yes	339	(10.4)	165	(48.7)	174	(51.3)	
**Invasive ventilation**							
No	2625	(80.9)	1399	(53.3)	1226	(46.7)	0.004
Yes	621	(19.1)	371	(59.7)	250	(40.3)	
**Death**							
No	2893	(89.1)	1559	(53.9)	1334	(46.1)	0.036
Yes	353	(10.9)	211	(59.8)	142	(40.2)	

* Deep venous thrombosis and pulmonary embolism; ** Acute myocardial infarction, myocardial ischemia, or coronary intervention. TIA, transient ischemic attack.

**Table 2 jcm-11-05951-t002:** Prevalence of myocardial injury * among patients who underwent troponin measurement, according to baseline characteristics.

Characteristics	Frequencies *n* (%)	Prevalence of Myocardial Injury *	Prevalence Ratio (95% CI)	χ^2^ Test *p*-Value
**Patients with measured troponin**	1476 (100)	23.9%		
**Sex**				
Male	895 (60.6)	25.5%	1.2 (0.97, 1.43)	0.081
Female	581 (39.4)	21.5%	1	
**Age**				
<40 years	254 (17.2)	7.5%	1	<0.001
40 to 59 years	609 (41.3)	14.0%	1.9 (1.16, 3.00)	
60 to 69 years	260 (17.6)	33.1%	4.4 (2.77, 7.04)	
≥70 years	353 (23.9)	46.2%	6.2 (3.95, 9.65)	
**Oxygen saturation**				
≥94%	1055 (71.5)	21.1%	1	<0.001
≤93%	421 (28.5)	30.9%	1.5 (1.21, 1.76)	
**Diabetes**				
No	1103 (74.7)	21.2%	1	<0.001
Yes	373 (25.3)	31.9%	1.5 (1.25, 1.81)	
**Hypertension**				
No	614 (41.6)	14.3%	1	<0.001
Yes	862 (58.4)	30.7%	2.1(1.72, 2.66)	
**Heart failure**				
No	1428 (96.7)	22.8%	1	<0.001
Yes	48 (3.3)	58.3%	2.6 (1.97, 3.30)	
**Coronary artery disease**				
No	1363 (92.3)	22.2%	1	<0.001
Yes	113 (7.7)	44.2%	2.0 (1.58, 2.50)	
**Chronic heart disease**				
No	1384 (93.8)	22.3%	1	<0.001
Yes	92 (6.2)	47.8%	2.1 (1.69, 2.71)	
**Chronic kidney disease**				
No	1418 (96.1)	22.8%	1	<0.001
Yes	58 (3.9)	50.0%	2.2 (1.67, 2.89)	

* Defined as troponin >1× upper reference limit. CI, confidence interval.

**Table 3 jcm-11-05951-t003:** In-hospital mortality and its relative risk according to baseline patient characteristics (*n* = 3246).

Characteristics	In-Hospital Mortality	Relative Risk (RR) (95% CI)	χ^2^ Test *p*-Value
**Troponin (×URL)**			
≤1 (negative)	5.5%	1	<0.001
>1 (myocardial injury)	22.7%	4.1 (3.01, 5.59)	
Not measured	11.9%	2.2 (1.64, 2.84)	
**Sex**			
Male	11.2%	1.1 (0.87, 1.30)	0.522
Female	10.5%	1	
**Age**			
<40 years	2.5%	1	<0.001
40–59 years	6.3%	2.5 (1.47, 4.32)	
60–69 years	12.6%	5.0 (2.92, 8.73)	
≥70 years	24.8%	9.9 (5.96, 16.71)	
**Oxygen saturation**			
>93%	7.9%	1	<0.001
≤93%	18.5%	2.3 (1.94, 2.86)	
**Diabetes**			
No	8.8%	1	<0.001
Yes	17.1%	1.9 (1.59, 2.37)	
**Hypertension**			
No	7.8%	1	<0.001
Yes	13.2%	1.7 (1.37, 2.11)	
**Heart failure**			
No	10.4%	1	<0.001
Yes	24.8%	2.4 (1.66, 3.38)	
**Coronary artery disease**			
No	10.4%	1	0.001
Yes	17.5%	1.7 (1.24, 2.29)	
**Chronic heart disease**			
No	10.4%	1	<0.001
Yes	20.4%	2.0 (1.43, 2.70)	
**Chronic kidney disease**			
No	10.2%	1	<0.001
Yes	25.0%	2.5 (1.83, 3.30)	
**Total**	10.9%		

CI, confidence interval; URL, upper reference limit.

**Table 4 jcm-11-05951-t004:** Multivariate analysis findings for the prediction of in-hospital mortality among patients with COVID-19.

Characteristics	Multiple Logistic Model		Final Logistic Model		
	aOR	*p*-Value	aOR	95% CI	*p*-Value
**Troponin (×URL)**		<0.001			<0.001
≤1 (negative)	1		1		
>1 (myocardial injury)	2.95	<0.001	2.99	2.06–4.36	<0.001
Not measured	2.17	<0.001	2.20	1.62–2.97	<0.001
**Sex**					
Male	1.20	0.149			
Female	1				
**Age (years)**		<0.001			<0.001
<40	1		1		
40–59	2.40	0.003	2.32	1.3 –4.07	0.003
60–69	4.36	<0.001	4.12	2.29–7.39	<0.001
≥70	9.64	<0.001	9.07	5.22–15.75	<0.001
**Oxygen saturation (%)**					
>93	1		1		
≤93	2.17	<0.001	2.17	1.71–2.75	< 0.001
**Diabetes**					
No	1		1		
Yes	1.37	0.018	1.35	1.05–1.73	0.018
**Hypertension**					
No	1				
Yes	0.85	0.260			
**Heart failure**					
No	1				
Yes	1.18	0.557			
**Coronary disease**					
No	1				
Yes	0.80	0.296			
**Chronic heart disease**					
No	1				
Yes	1.19	0.467			
**Kidney disease**					
No	1				
Yes	1.39	0.128			

aOR = adjusted odds ratio; CI, confidence interval; URL, upper reference limit.

**Table 5 jcm-11-05951-t005:** Probability of death according to the final logistic regression model (*n* = 3246).

Age (Years)	Diabetes	O_2_ Sat	Troponin		
			≤1 × URL (No Injury)	>1 × URL (Injury *)	Not Measured
<40	No	>93%	1.2%	3.4%	2.5%
		≤93%	2.5%	7.1%	5.3%
	Yes	>93%	1.6%	4.3%	3.4%
		≤93%	3.3%	9.3%	7.0%
40–59	No	>93%	2.6%	7.5%	5.6%
		≤93%	5.6%	15.0%	11.5%
	Yes	>93%	3.5%	9.9%	7.4%
		≤93%	7.4%	19.2%	14.9%
60–69	No	>93%	4.6%	12.6%	9.6%
		≤93%	9.5%	23.8%	18.7%
	Yes	>93%	6.1%	16.3%	12.5%
		≤93%	12.4%	29.7%	23.7%
≥70	No	>93%	9.6%	24.1%	18.9%
		≤93%	18.7%	40.8%	33.6%
	Yes	>93%	12.5%	30.0%	23.9%
		≤93%	23.7%	48.2%	40.6%

* Myocardial injury, defined as troponin >1× upper reference limit. O_2_ sat, oxygen saturation, URL, upper reference limit.

## Data Availability

The data presented in this study are available on request from the corresponding author.

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
