# Peer review of "Troponin in COVID-19: To Measure or Not to Measure? Insights from a Prospective Cohort Study"

_jcm, 2022, doi:10.3390/jcm11195951_

Round 1
Reviewer 1 Report
Dear Editor,
Moll-Bernardes presented the results of their prospective study on the possible prognostic value of Troponin in patients with COVID-19. This is a very nice work. I would like to comment the authors and would also like to thank you for this opportunity.
Please see my comments below.
Introduction:
*This is a well-written section. The objective is clearly defined.
*I would like to kindly ask the authors to review again the literature on the prevalence of troponin elevation in hospitalized patients with COVID-19. My clinical experience and anecdotal data from my institution suggest that the prevalence is much lower than 20%. May authors re-review literature and additionally cite studies that report lower prevalence (if any)?
*Pulmonary embolism is another acute non-ischemic injury. Pulmonary embolism has been associated with COVID-19 (https://journals.sagepub.com/doi/full/10.1177/0268355520955083)
Methods:
*This is a well-written section overall.
*Inclusion and exclusion criteria need to be defined clearly.
*Why did researchers calculate RR for the primary outcome and OR for in the logistic regression
Results:
*I would define thromboembolic phenomena.
*I would always provide CI while reporting effect size estimates
*Technically, OR refers to odds (likelihood) and RR refers to risk.
*It is a limitation of this study that obesity was not taken into account. Obesity is possibly the strongest risk for factor for worse outcomes in COVID-19 with older age and male sex.
*A supplementary diagram showing the flow of patients from initial cohort to the final cohort with reasons for exclusion would help.
Discussion:
*I would simplify this section and would focus on the main finding that is the possible prognostic value of Troponin.
*I would try to discuss how this finding can be useful in clinical practice. How troponin measurement in the absence of suspicion for ACS, acute PE, acute aortic dissection, or myocarditis would change our management? What can this test offer that our current prognostic tools cannot do?
Author Response
Responses to Reviewer #1 comments
Introduction:
*This is a well-written section. The objective is clearly defined.
Response (R.) We thank the reviewer for this nice comment.
*I would like to kindly ask the authors to review again the literature on the prevalence of troponin elevation in hospitalized patients with COVID-19. My clinical experience and anecdotal data from my institution suggest that the prevalence is much lower than 20%. May authors re-review literature and additionally cite studies that report lower prevalence (if any)?
- We appreciate this important observation. As suggested, we revised again the literature and found, among other studies, a meta-analysis (1) published in 2021, which reported an incidence rate of troponin elevation ranging from 7% to 44%. This information was revised in the text and the reference was included (line 44).
Reference: (1) Aikawa T, Takagi H, Ishikawa K, Kuno T. Myocardial injury characterized by elevated cardiac troponin and in-hospital mortality of COVID-19: An insight from a meta-analysis. J Med Virol. 2021 Jan;93(1):51-55. doi: 10.1002/jmv.26108. Epub 2020 Jun 19. PMID: 32484975; PMCID: PMC7301031.
*Pulmonary embolism is another acute non-ischemic injury. Pulmonary embolism has been associated with COVID-19 (https://journals.sagepub.com/doi/full/10.1177/0268355520955083)
- We appreciate this comment, and we understand that Pulmonary Embolism must be considered in the analysis as a possible cause of myocardial injury, particularly in the presence of right ventricular dysfunction. Pulmonary Embolism was included as a secondary outcome together with deep venous thrombosis (named as thromboembolic phenomena). We have included this information in the text (line 108).
Methods:
*This is a well-written section overall.
- We thank the reviewer for this kind comment.
*Inclusion and exclusion criteria need to be defined clearly.
- Thank you for the excellent suggestion. The study was part of a multicentric registry, so all patients with confirmed COVID-19 diagnosis were included consecutively. In addition, a substantial number of patients who were initially included in the registry, were excluded from the analysis because some centers did not have enough research personal to collect clinical data during some turbulent periods of the pandemic, or because SARS-CoV2 diagnosis was not confirmed. Missing data pattern was completely at random, and unrelated to comorbidities, outcomes or to myocardial injury. A flow diagram (Figure S1) was included in the Supplementary Material with this information, and we included a brief discussion of patients’ exclusion in study limitations (lines 332-335).
*Why did researchers calculate RR for the primary outcome and OR for in the logistic regression
- We appreciate this interesting question regarding our choices of association measures. In the initial exploratory analysis, we chose to use RR as we believed it would allow an easier interpretation of the impact of each variable. Nevertheless, for the multiple analysis we opted for a logistic regression model, and for this reason we used OR. We could include OR in the univariate analysis as well, if the revisor recommends it, however we tried to avoid including too much information and numbers to enable an easier understanding of our analysis. We included 3 lines in Table 3 with in-hospital mortality and RR according to troponin levels (normal, increased and not measured) to improve understanding.
Results:
*I would define thromboembolic phenomena.
- Thank you for this observation. As mentioned, we included pulmonary embolism and deep venous thrombosis as “thromboembolic phenomena” in our secondary outcomes. This information is now included in the text (lines 107-108).
*I would always provide CI while reporting effect size estimates
- We thank the reviewer for this important suggestion. We understood that by “effect size” the reviewer is referring to measures of association (Prevalence Ratio and RR), so we included all confidence intervals in Table 2 (for Prevalence Ratio estimates) and in Table 3 (for Relative Risk estimates).
*Technically, OR refers to odds (likelihood) and RR refers to risk.
- As we mentioned previously in the item concerning RR and OR, we agree with the concepts so accurately pointed out by the reviewer, and we understand the fact that, although these measures are associated, they are different. In Table 3 we used RR (instead of OR) as our aim was to describe the findings to be easily interpreted.
*It is a limitation of this study that obesity was not taken into account. Obesity is possibly the strongest risk for factor for worse outcomes in COVID-19 with older age and male sex.
- We are thankful for this important observation. We understand that obesity is considered an important risk factor in COVID-19, however, we did not have good quality information regarding BMI to include this comorbidity in our analysis. We believe the lack of this information for many patients was due to the large number of patients hospitalized during the pandemic, associated with a reduced number of health care providers and increased work burden for some periods. We included this limitation in the discussion (lines 330-332).
*A supplementary diagram showing the flow of patients from initial cohort to the final cohort with reasons for exclusion would help.
- We appreciate the reviewer’s suggestion, and a flow diagram of patient recruitment was included in Supplementary Material as Figure S1.
Discussion:
*I would simplify this section and would focus on the main finding that is the possible prognostic value of Troponin.
- We understand that some parts of the discussion were not very clear. We are grateful for this suggestion that gave us the opportunity of better explaining our findings.
*I would try to discuss how this finding can be useful in clinical practice. How troponin measurement in the absence of suspicion for ACS, acute PE, acute aortic dissection, or myocarditis would change our management? What can this test offer that our current prognostic tools cannot do?
- We thank the reviewer for this comment. Our findings suggest that a negative troponin result may indicate a better prognosis and a higher probability that a patient will not need intensive support, or might have an earlier discharge from the hospital, permitting a better resource allocation. Patients with higher troponin levels must be surveilled closely, may need intensive care or other invasive support. Patients with no troponin measurement might have a better risk prediction if we measured this biomarker, as the lack of measurement was associated with an OR of 2.2, compared to patients with no myocardial injury, after adjusting for age, sex, oxygen saturation and other comorbidities. These findings show that troponin is an independent risk predictor and suggest that troponin should be routinely measured in patients hospitalized with COVID-19. We have included these points in our discussion.

Reviewer 2 Report
Troponin in COVID-19: To measure or not to measure
Insights from a prospective cohort study
An interesting study of outcomes among 4268 COVID-19 infected admitted patients where 3246 patients were included in this cohort study. Troponin was taken the first week of hospitalization on 1476 patients. Myocardial injury was detected in 353 patients and the study mainly is concerning the prognostic implications of an elevated troponin level.
There are some questions that may need some attention:
1. Please precisely state how MI was defined. The assay specific URL?
2. Please make a table with the troponin assays that were in use, how many patients that were analyzed with these troponin assays, the frequency above URL as an indication of the MI frequency depending on the troponin-assay used
3. The table 1 could be simplified and even put in the Supplementary. The current version is very hard to read and the information it analyzes really unclear. Why do you, or do not, choose to analyze troponin?
4. I might have missed this, but could the authors state the median number of troponin tests per patient that were analyzed in their study.
5. In table 3 the authors need to clarify if the percentages are from the entire study cohort or only the ones with a troponin measurement?
6. Please analyze if there is any significant difference in sex, age and what you have complete information on in the excluded patients due to incomplete data.
7. Please define what is meant by "incomplete data" and what this exclusion criteria really was based on. This is to limit problems of possible selection bias in the study.
8. I like figure 4. Gives a lot of information.
9. The last part, the logistic regression is something that I do not understand fully, perhaps because the statistical analysis is not very well described. Please make sure that most readers understand what you have done, and not done, in this analysis. For instance, one usually lists the parameters that one has used to make the regression model in the table. I also do not understand how troponin can be part of the model if all patients in the study, among which only 1476 had a troponin measured. The authors should clearly state that only the 1476 patients with a troponin measured were used in the regression and building the model. The table 5 states that the probability of death was calculated based on 3246 patients. Typo? How can one make a model using troponin as a factor when patients without a troponin measurements are included.
